# Custom-made artificial eyes using 3D printing for dogs: A preliminary study

**So-Young Park**[1], **Jeong-Hee An**[1], **Hyun Kwon**[1], **Seo-Young Choi**[1], **Ka-Young Lim**[1], **Ho-Hyun Kwak**[2], **Kamal Hany Hussein**[2,3], **Heung-Myong Woo**[2], **Kyung-Mee Park**[1]*

1 Department of Ophthalmology and Surgery, College of Veterinary Medicine, Chungbuk National University, Cheongju, Korea, 2 Department of Surgery, College of Veterinary Medicine, Kangwon National University, Chuncheon, Korea, 3 Department of Animal Surgery, College of Veterinary Medicine, Assiut University, Assiut, Egypt

* parkkm@cbnu.ac.kr

## Abstract

Various incurable eye diseases in companion animals often result in phthisis bulbi and eye removal surgery. Currently, the evisceration method using silicone balls is useful in animals; however, it is not available to those with impaired cornea or severe ocular atrophy. More-over, ocular implant and prostheses are not widely used because of the diversity in animal size and eye shape, and high manufacturing cost. Here, we produced low-cost and custom-ized artificial eyes, including implant and prosthesis, using computer-aided design and three-dimensional (3D) printing technique. For 3D modeling, the size of the artificial eyes was optimized using B-mode ultrasonography. The design was exported to STL files, and then printed using polycaprolactone (PCL) for prosthesis and mixture of PCL and hydroxy-apatite (HA) for ocular implant. The 3D printed artificial eyes could be produced in less than one and half hour. The prosthesis was painted using oil colors and biocompatible resin. Two types of eye removal surgery, including evisceration and enucleation, were performed using two beagle dogs, as a preliminary study. After the surgery, the dogs were clinically evaluated for 6 months and then histopathological evaluation of the implant was done. Ocular implant was biocompatible and host tissue ingrowth was induced after *in vivo* application. The cus-tom-made prosthesis was cosmetically excellent. Although long-term clinical follow-up might be required, the use of 3D printed-customized artificial eyes may be beneficial for ani-mals that need personalized artificial eye surgery.

## Introduction

Various incurable eye diseases in dogs often require eye removal surgery, with poor aesthetic results of facial deformation [1–3]. However, many dog owners prefer surgical procedures that can preserve the eye shape, despite the lack of vision. Therefore, evisceration using intraocular silicone prosthesis (ISP) as an orbital implant has been performed frequently in dogs [4, 5]. Inserting an ISP after evisceration has cosmetically acceptable results; however, it has some limitations. This method cannot be used in patients with corneal problems or severe ocular atrophy [6]. Moreover, corneal opacity, vascularization, fibrosis, and pigmentation may be induced after surgery [7–10]. Although 62% of the owners whose dogs underwent both

**Funding:** This work was supported by the Global Research and Development Center (GRDC)

Program through the National Research Foundation of Korea (NRF) funded by the Ministry of Education, Science and Technology (2017K1A4A3014959) and NRF-2018R1D1A1B07050014 (https://www.nrf.re.kr). This work was also supported by the research grant of the Chungbuk National University in 2017 (https://www.chungbuk.ac.kr). The funders had no role in study design, data collection and analysis, decision to publish, or preparation of the manuscript.

**Competing interests:** The authors have declared that no competing interests exist.

evisceration and ISP implantation were satisfied with the outcome, others showed dissatisfaction owing to complications, including corneal erosion, orbital inflammation, and dehiscence of the surgical site [11, 12].

In humans, most patients wear artificial eyes after eye removal surgery. The artificial eye currently being used in humans consists of two components. First, the orbital implant is placed in the orbit during eye removal surgery. It is used to replace the orbital volume to prevent facial deformation and improve the movement of the ocular prosthesis by maintaining the movement of the extraocular muscles. Porous orbital implants made of hydroxyapatite (HA) or polyethylene (PE) are preferred to non-porous glass, silicone, or acrylic (polymethylmethacrylate, PMMA) implants as the former allow fibrovascular ingrowth [13–15]. In contrast, porous implants are not commonly used in animals because they are much more expensive than silicone balls. The second component of the artificial eye is the ocular prosthesis, which is similar in appearance to the fellow eye on the opposite side and can move like the normal eye [16–18]. Currently, custom-made prosthesis (CMPs) made of PMMA are mainly used in humans because they are more comfortable and aesthetically more acceptable than ready-made prosthesis [19, 20]. Their use in small animals is relatively uncommon although they have shown good results, with few major complications and better appearance [6, 21]. This is because CMPs are very expensive, take a long production time, and are labor-intensive [4]. In addition, they are manufactured by skilled technicians in several stages and require frequent patient visits during manufacture [21]. The significant variation in ocular shape according to dog breed and size is also one of the hurdles to ready-made artificial prosthesis use.

The main strength of 3D printing is the ability to make products conveniently and cost-effectively [22–24]. Currently, 3D printed bio-models are widely used in the medical field because they can provide tactile feedback and reproduce anatomical structures and movements [25–27]. The 3D printing technology is used not only for planning surgical procedures, making intraoperative guidance devices, and training but also for producing facial prostheses of the nose and ear [28–31]. Herein, we aimed to create custom-made artificial eyes including ocular implants and prosthesis for dogs using 3D printing and evaluate their clinical performance in order to overcome these challenges. The 3D printed customized artificial eyes might be beneficial medical tools in veterinary clinics for many companion animals that need to undergo eye removal surgery owing to incurable eye diseases.

## Materials and methods

### Clinical examination

The size of the orbital implant was made to be 75% of the ocular volume by measuring the longest diameter of the eyeball using B-mode ultrasonography, according to a previous report [32]. For size of ocular prosthesis, the X-axis and Z-axis were determined by the diameter of the eyeball that passes through the transverse section of the lens and length from the cornea to the posterior lens capsule in the transverse plane of the ultrasonography, respectively (Fig 1). The Y-axis was decided as the distance from the superior conjunctival fornix to the inferior conjunctival fornix. If prosthesis had a short Z-axis, it was not fit on the conjunctival surface due to its low curvature. Based on the ultrasound, we determined the size of the implant as 20 x 20 x 20 mm for both dogs. In case of the implant and conformer, X, Y, and Z axis were 19.7, 20, and 10.5 mm for dog 1 and 21.1, 10.8, and 10.3 mm for dog 2, respectively.

### Computer-aided design

Based on the results, artificial eyes including orbital implant, conformer, and ocular prosthesis were designed using a computer-aided design (Thinkercad, Autodesk, Mill Valley, CA, USA).

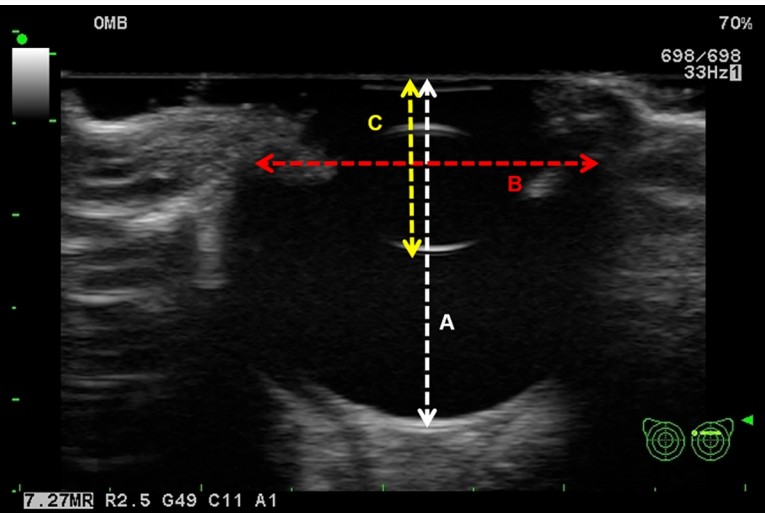

**Fig 1. Size optimization of ocular implant and prosthesis using ultrasonography.** For production of the ocular implant, the longest diameter of the eyeball was measured (A, white dot line). The implant size along the X, Y, and Z axes was determined to be 75% of its length. For the prosthesis, the length along the X-axis was the length of the eye that passes through the diameter of the lens in the transverse plane of the ultrasound (B, red dot line). The length along the Z-axis was measured from the cornea to the posterior lens capsule (C, yellow dot line).

This 3D model designs were saved as a stereolithography file and then converted to a G-code file (Creator K, Rokit Healthcare) for printing (Fig 2).

## Material preparations for modeling

Polycaprolactone pellet (PCL; average molecular weight: 45,000; Sigma-Aldrich, St. Louis, MO, USA) and 40% w/v HA powder (particle size < 200 nm, Sigma-Aldrich) mixtures were used for developing the orbital implant. PCL/HA was dissolved in methylene chloride (Duksan, Ansan, Korea) and stirred in the glass beaker. Subsequently, PCL/HA solution was applied thinly to the beaker wall and dried overnight at room temperature. When the PCL/HA solution was completely solid, it was cut small enough to fit into the barrel of the 3D printer (INVIVO, Rokit Healthcare, Seoul, Korea). For conformers and ocular prosthesis, the PCL filament (Rokit Healthcare) was used.

## Evaluation of microstructure of ocular implant

To confirm whether HA nanoparticles were evenly printed on the PCL polymer, a field emission scanning electron microscope (FE-SEM, Ultra Plus, Carl Zeiss, Oberkochen, Germany)

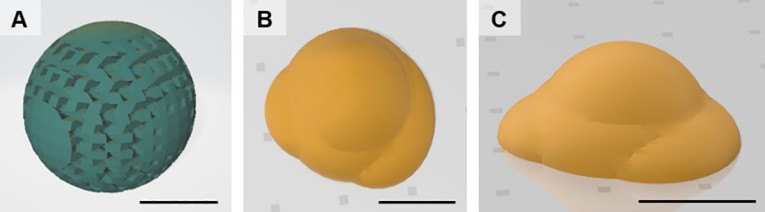

**Fig 2. Three-dimensional model designed using computer-aided design.** Orbital implant (A), Ocular prosthesis; sky view (B) and side view (C). Bar = 1 cm.

was used. Various ratio of HA and PCL mixture was examined. Platinum coating was employed prior to examination.

## Experimental animals

The animal experiments were approved by Chungbuk National University Animal Care and Use Committees (Number: CBNUA-1155-18-01) of Laboratory Animal Research Center at Chungbuk National University (Cheongju, Korea). To check the *in vivo* applicability of our 3D printed artificial eyes using two surgical methods including evisceration and enucleation, two beagle dogs were preliminary tested. Two healthy conventional 2-year-old beagle dogs (one male, 8.4kg and one female, 6.4kg) were obtained from DooYeol Biotech (Seoul, Korea) and randomly selected for surgery. The dogs were acclimated for 2 weeks at Laboratory Animal Research Center at Chungbuk National University before experiments and maintained following conditions: an ambient temperature of $20 \pm 2°C$, relative humidity of $50 \pm 10\%$, air ventilation rate of 10 cycles/h, and a 12:12 h light:dark cycle. The dogs were fed a commercial dry food (Pro-Plan SPORT, Purina, St. Louis, MO) mixed with canned food (a/d$^{TM}$ Canine/ Feline critical Care, Hills Pet Nutrition, Topeka, KS) twice a day. The dogs were freely accessed to fresh water during all experimental periods. The animal care staffs checked the health and well-being of the animals by performing physical examination and monitoring animal's behaviors every day. Complete blood cell count, serum chemistry, and microbial test were performed before and after surgery. For environmental enrichment, exercising was done by animal care staffs and dog toys and chews were provided every day. The surgeries were done at animal surgical facilities of Chungbuk National University. We evaluated the post-operative prognosis for 6 months. After that, the dogs were euthanized for the histological examinations.

## Surgical approaches

All dogs were fasted for at least 8 h before anesthesia, but had free access to water. General anesthesia was performed using acepromazine (0.05mg/kg IV; Sedaject, Samu Median, Seoul, Korea) for premedication, alfaxalone (1.5 mg/kg IV; Alfaxan, Jurox, Rutherford, Australia) for induction, and isoflurane (Terrell; Piramal Critical Care, Bethlehem, PA) for maintenance at 2–3% in a semi-closed circuit system. After anesthetic stabilization of the dogs, atracurium (0.2mg/kg IV; Atra Inj, Hana Pharm, Seoul, Korea) was injected. The eyeballs and eyelid were disinfected with 0.2% and 2% povidone iodine solution, respectively. After that, retrobulbar anesthesia using 1ml of 2% lidocaine solution (Jeil Pharmaceutical, Daegu, Korea) was done. The tidal volume was controlled at 10-20ml/kg with 10–15 respiratory rates/min using ventilating system. During anesthesia, heart rate, oxygen saturation, end-tidal $CO_2$, respiratory rate, blood pressure, blood glucose concentration, and body temperature were monitored continuously by veterinarians.

**Eye removal surgery 1: Evisceration.** For eye removal, dog 1 (male) underwent evisceration surgery on the left eye by the four petal technique (S1 Fig). Briefly, a 360-degree conjunctival peritomy was performed, and sub-Tenon's blunt dissection was conducted (S1A Fig). After the limbal stab incision, first the corneal button and then the ocular contents were removed (S1B Fig). The remaining uveal tissue was removed using a sterile gauze, and the internal scleral surface was washed with sterile normal saline. Four sclerotomies were performed between the rectus muscle insertions from the limbus to the optic nerve to make 4 petals (S1C Fig). Each petal was released from the optic nerve, and the orbital implant was placed in 4 petals (S1D Fig). The 2 petals were sutured vertically to each other using interrupted 6–0 polydioxanone (PDS) sutures in front of the orbital implant without tension. Subsequently, the other 2 petals were sutured in the same manner over the vertical petals (S1E Fig). Tenon's

capsule and conjunctiva were sutured with 6–0 interrupted PDS sutures (S1F Fig). Finally, the third eye lid was removed, and then, the conformer was inserted and temporal tarsorrhaphy was performed (S1G and S1H Fig).

**Eye removal surgery 2: Enucleation.** Dog 2 (female) underwent enucleation surgery on the left eye by the myoconjunctival technique (S2 Fig). Briefly, under general anesthesia, an eyelid speculum was placed between the eyelids. A lateral canthotomy was performed. Furthermore, a 360-degree peritomy was performed at the limbus (S2A Fig). Anterior Tenon's capsule was separated from the sclera. Each of the rectal muscles was identified, hooked, and double tied, first with 4–0 silk suture just short of the muscle insertion, and then with 6–0 double-armed PDS suture, about 6 mm distally (S2B and S2C Fig). Each rectal muscle was transected at a point between the two sutures; 4–0 silk sutures serve as traction sutures, while 6–0 PDS sutures will later be used to suture the muscles through the conjunctiva. Superior oblique and inferior oblique muscles were hooked, transected, and allowed to retract posteriorly. With a gentle forward traction on the eyeball using the four silk sutures, the optic nerve was palpated with the closed tip of the scissors (S2D Fig). The nerve was transected with one bold cut, and hemostasis was achieved by pressure. The orbital implant was placed posterior to posterior Tenon's capsule (S2E Fig). The posterior Tenon's capsule was closed with interrupted 6–0 PDS sutures (S2F Fig). Each rectal muscle was sutured through the anterior Tenon's capsule and conjunctiva using preplaced double-armed PDS sutures. Anterior Tenon's was closed with interrupted 6–0 PDS sutures. After removing the third eye lid, conjunctival closure was performed with continuous 6–0 PDS suture. The conformer was placed, and temporal tarsorrhaphy was performed (S2G and S2H Fig).

## Pre- and post-operative phase

For pre-operative care, intravenous fluid (H/S, CJ Health Care, Seoul, Korea) was supplied at a rate of 5 ml/kg/h from 3 hours before operation and for a total of 8 hours. Before surgery, ampicillin-sulbactam (22 mg/kg; Sulbacin inj, Dongkwang, Seoul, Korea), famotidine (0.5 mg/kg, Dong-A Pharm, Seoul, Korea), and tramadol (4mg/kg; Tridol inj, Yuhan, Seoul, Korea) were injected intravenously.

For post-operative care, for the first three days, ampicillin-sulbactam (22 mg/kg IV, q12h), famotidine (0.5 mg/kg IV, q12h), prednisolone (1mg/kg SC, q24h; Soron, Handong, Seoul, Korea), and tramadol (4 mg/kg IV, q 8h) were injected. After that, amoxicillin-clavulanate (22 mg/kg; Amocra, Kuhnil, Seoul, Korea), prednisolone (0.5 mg/kg; Sorondo, Yuhan), famotidine (0.5 mg/kg, Nelson, Seoul, Korea), and tramadol (4mg/kg; Tridol, Yuhan) were used per orally q12h for the first 2 weeks after surgery, and subsequently, prednisolone was gradually tapered in the next 2 weeks. As topical eye drops, ofloxacin (OcuFlox; Samil, Seoul, Korea) and prednisolone (Pred Forte; Allergan, Dublin, Ireland) were administered q8h for the first 2 weeks and q12h for the next 2 weeks. The tarsorrhaphy was removed 2 weeks postoperatively. The conformer was worn for 6 weeks after surgery and then replaced with prosthesis.

## Management of the prosthesis

For *in vivo* application, sterilization using ethylene oxide (EO) gas was performed one day before operation. We found that the shape of the implant and conformer were deformed after autoclaving or plasma sterilization; however, EO gas, which does not generate heat, did not induce deformation. However, after coloring and coating of the prosthesis, the painted part collapsed after EO gas sterilization. Therefore, the painted prosthesis was disinfected by soaking into slightly acidic electrolyzed water (SAEW; Medilox, Soosan, Seoul, Korea) for more than 10 minutes at room temperature.

For daily management of prosthesis after operation, the prosthesis was disinfected with SAEW or commercial contact lens cleaner and then rinsed with sterile saline before applying. We found that neither the paint faded nor did the shape change upon soaking of the prosthesis in SAEW for more than 6 months.

### Evaluation of outcomes

To assess clinical applicability, we checked gloss appearance, inflammation, implant extrusion, discharge amount, infection, and pain after the surgery for 6 months. The grade of appearance was subjectively classified into five categories by three veterinarians: poor, bad, fair, good, and excellent. The level of discharge and pain was subjectively classified into four categories by three veterinarians: severe, moderate, mild, and none.

To check the *in vivo* host responses or biocompatibility 6 months after implant insertion, the dogs were euthanized using 10ml of T-61 euthanasia solution (Merck Animal Health, NJ) under general anesthesia as we mentioned above and then the implants were harvested. The harvested tissues were fixed in 10% phosphate buffered formalin and embedded in paraffin. Sections were cut at 4um thickness and stained with hematoxylin and eosin.

## Results

### Optimization for printing conditions

First, we established the optimal printing conditions for artificial eyes as described S1 Table. Appropriate optimization of printing conditions, such as nozzle size and output speed, was important to accurately print the desired design. For output of the orbital implant, the mixture of PCL/HA was placed into the barrel of the air dispenser before printing. The air dispenser temperature was set to 100˚C to melt the PCL/HA for 12 hours. For printing, a 400-μm nozzle was used. In our preliminary study, 200-μm nozzles clogged frequently during printing, while the 600-μm nozzle did not clog, but the accuracy was poor. We set the bed temperature to 10˚C to allow for quick solidification of PCL/HA; otherwise the material flowed down due to gravity at the time of implant output. Finally, we successfully produce customized porous PCL/HA ocular implant (Fig 3A and 3B).

Next, the conformer and prosthesis were fabricated using PCL filament. For output of the conformer and ocular prosthesis, PCL filament was mounted on the extruder heated at 130˚C

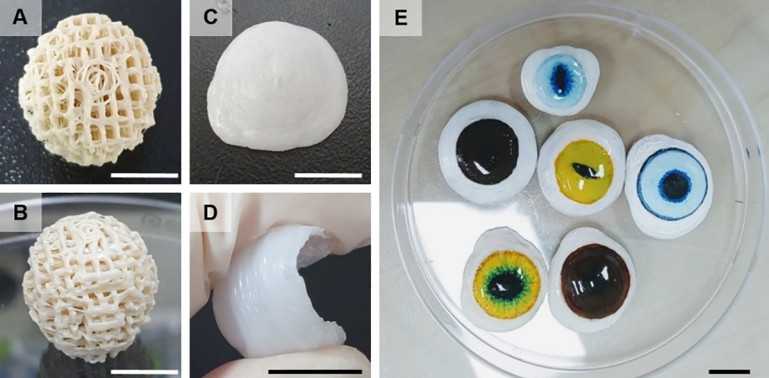

**Fig 3. Three-dimensional printout of the orbital implant.** Sky view (A) and side view (B); ocular prosthesis, sky view (C) and flexible side view (D). The prosthesis was painted with an acrylic color similar to the intact eye. After drying, it was coated with a biocompatible light-curing varnish (E). Bar = 1cm.

for 3 minutes. In our preliminary study, we had set a mesh-like support structure to fill the internal hollow of the prosthesis and conformer in order to prevent the prosthesis from sinking down during the printing. However, removing the support structure was difficult and the internal surface became rough. Therefore, we optimized the output condition as described above, and consequently the PCL filament was well printed without the support structure. The material did not sink down and piled up well; moreover, the surface was smooth (Fig 3C). We set the thickness of the conformer and prosthesis as 200μm because they were flexible at this thickness, which allowed for ease in wearing and removal (Fig 3D). When the prosthesis was thicker, it was harder and difficult to wear. After printing, two holes were drilled using a 4-mm biopsy punch for making the conformer. Finally, we could produce CMP for companion animals using 3D printing in a short time. In the case of the ocular prosthesis, it was colored with acrylic paint similar to the opposite eye. Thereafter, the prosthesis was coated with a biocompatible light-curing resin (Fotoplast® Lack M, Dreve, Unna, Germany) and cured for 5 minutes under 365-nm ultraviolet light. (Fig 3E)

The 3D prosthesis printing procedure takes 30 minutes. For printing the ocular implant, it takes 50 minutes. The total production cost of our artificial eyes was approximately about $ 26 for an implant and $ 1.6 for a prosthesis, respectively. In addition, it is easy to re-produce without skilled ocularists because design data for each patient could be stored.

## Evaluation of microstructure of ocular implant

To confirm that the PCL and HA were mixed evenly in the implant, the surface microstructure of PCL/HA composite scaffolds was observed in FE-SEM images. We found that HA nanoparticles were spread evenly in the PCL frames after printing. (Fig 4)

## The 3D printed artificial eye was clinically available and biocompatible

**Fitting the prosthesis.** The implant size was adequate during surgery, and enophthalmos did not occur after surgery. For regular management, the painted prosthesis was worn for 10 hours three days a week. Local anesthesia was not needed while applying and removing the prosthesis, because its flexibility allowed for easy application and removal.

The prosthesis was evaluated with regard to color and symmetry as compared to the opposite eye. The grade of appearance in two dogs was rated as good to excellent because the prosthesis was very similar in appearance to the intact eye (Fig 5).

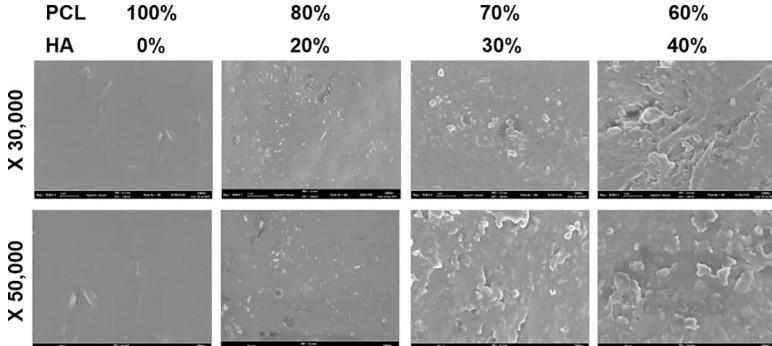

**Fig 4. Analysis of microstructure of polycaprolactone (PCL)/hydroxyapatite (HA) composite using scanning electron microscope for production of the ocular implant.** As the percentage of HA increased, more particles were seen. Even distribution of HA nanoparticles in the PCL scaffold was seen, indicating that our printing method was suitable for implant production.

**Evisceration** **Enucleation**

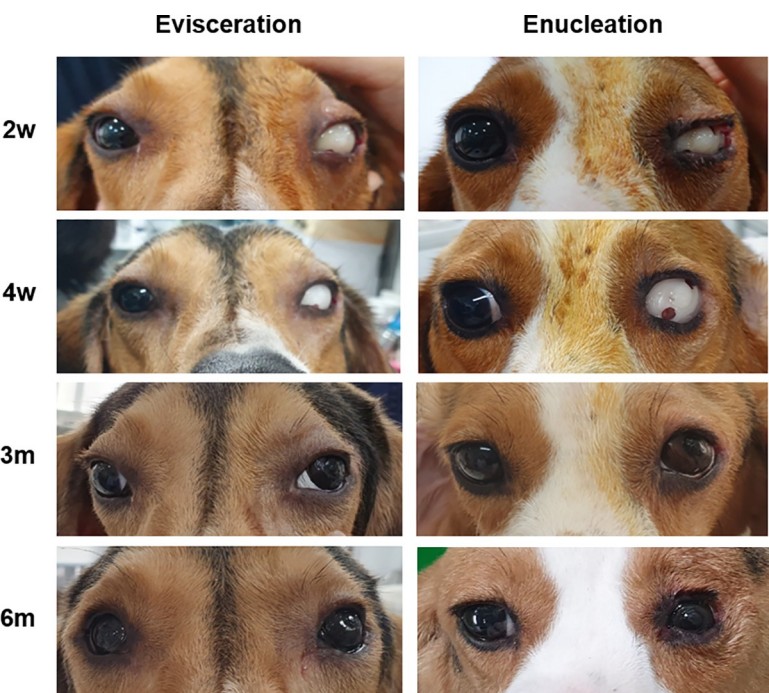

**Fig 5.** Postoperative gross findings 2 and 4 weeks after applying ocular conformer (A and B, respectively) in eviscerated or enucleated dogs. Six weeks after surgery, the conformer was removed, and prosthesis was applied (C). The appearance of the dog wearing prosthesis was good to excellent after more than six months.

**Dog's perception.** The degree of pain was measured, and until the third day after the surgery, moderate pain was detected. However, the pain disappeared within 2 weeks after operation in both dogs (Table 1).

The level of ocular discharge was mild to moderate in dog 1 and dog 2 until 6 months after surgery. The quantity of tear, as measured by the Schirmer tear test, was decreased significantly in two dogs after the surgery (Dog 1: pre-op 25 mm/min vs. post-op 11 mm/min, Dog 2: pre-op 19 mm/min vs. post-op 4 mm/min).

**Post-operative evaluation.** No clinical complications such as inflammation, infection, or extrusion of the ocular implant were observed during the 6 months after surgery (Fig 6A, a). Moreover, on the gross findings of harvested implants from the sacrificed dogs, host tissue formation within porous structure was shown (Fig 6A, b and c). In histology, fibro-vascular tissue

**Table 1. Post-operative clinical signs after ocular implant application for 6 months.**

| Clinical signs | 1d | | 7d | | 1m | | 3m | | 6m | |
|---|---|---|---|---|---|---|---|---|---|---|
| | EV | EN | EV | EN | EV | EN | EV | EN | EV | EN |
| Pain | ++ | ++ | + | ++ | - | - | - | - | - | - |
| Eyelid swelling | ++ | - | - | - | - | - | - | - | - | - |
| Extrusion | - | - | - | - | - | - | - | - | - | - |
| Ocular discharge | +++ | +++ | ++ | ++ | + | ++ | + | ++ | + | ++ |

EV: Evisceration.

EN: Enucleation.

d:day, m:month.

-: None, +: Mild, ++: Moderate, +++: Severe.

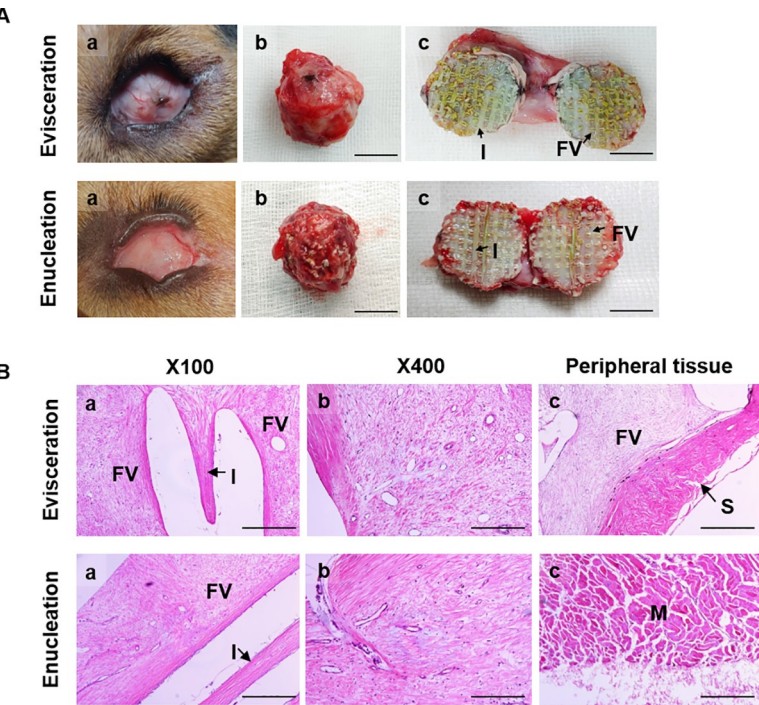

**Fig 6. Postoperative findings 6 months after surgery.** The results showed no evidence of inflammation and rejection (A, a). The morphology of implants were well maintained (A, b), moreover, fibro-vascular tissue (FV) ingrowth from the host was formed into the implants (I) (A, c). Bar = 1cm. In histology, fibro-vascular tissue was shown in the porous structure of the implants (B, a and b). Numerous connective tissues and vessels were shown. Also, no inflammation and rejection was shown in the peripheral tissues such as sclera (S) and muscle (M) near the implants (B, c). Bar = 100μm.

ingrowth was shown into the implant (Fig 6B, a and b). In addition, no inflammation occurred both in the implant and peripheral tissues in both dogs (Fig 6B, a-c). These results demonstrated that 3D printed artificial eyes including prosthesis and implant are not only clinically useful but also biocompatible.

## Discussion

In the present study, we produced artificial eyes including an implant and prosthesis for dogs using 3D printing. 3D printing has proven its potential in medical areas, such as pre-operative planning, intraoperative guidance, and fabrication of prostheses. The 3D printed bio-models are useful in a variety of surgical fields such as maxillofacial, orthopedic, cardiothoracic, and vascular surgeries [33–35]. Recently, the use of 3D printing in the veterinary medicine is also expanding; however, there is no report regarding the ophthalmologic application.

We found that application of 3D printing allows to decrease the production time, price, and the labor for the production of the artificial eyes. Conventional CMP production takes several days and needs two to four visits [36]. In contrast, our 3D prosthesis printing procedure takes only 30 minutes for printing and do not need several visits. For printing the ocular implant, it only takes 50 minutes. Commercially available HA implants for human are very expensive, usually more than $ 1000, so it could be a burden to animal owners. Here, the total production cost of our artificial eyes was approximately about $ 26 for an implant and $ 1.6 for a prosthesis, respectively. In addition, it is easy to re-produce without skilled ocularists because design data for each patient could be stored.

Recently, production of CMP for humans using the 3D printing technology have been progressing. In a previous report, wax model was produced by computed tomography (CT) and then fabricated the prosthesis by 3D printing [36]. However, it still needs the manual impression process for the wax model as the conventional method. As the initial approach, there were no big differences from traditional methods by making acrylic mold directly from the wax model. In addition, the soft tissue distortion of the anophthalmic socket could occur owing to the pressure of the molding material. To overcome these limitations, a prosthesis was created by scanning the anophthalmic socket using CT, without creating an impression mold in the conventional way [37]. However, the prosthesis itself must be made manually in the conventional way. In addition, it could be a burden for animal owners in terms of the cost for CT examination, including general anesthesia. Therefore, in our study, we optimized a sizing method using ocular ultrasound instead of CT scanning for reducing the burden of owners on anesthesia and CT costs. More recently, prostheses were produced using 3D printing more accurately [38]. A digital light processing (DLP) printer with liquid materials was used with high precision and output; however, the disadvantage is that the equipment and materials are expensive. In our study, we used a fused deposition modeling (FDM) 3D printer, which has a lower precision than the DLP printer; however, the equipment and materials are more economic [39]. In dogs, the need for accurate coloring is less important than in humans, which could result in satisfactory results with an FDM printer.

The production of ocular implants using 3D printing was first reported in our study. To fabricate the orbital implant, we used a mixture of PCL and 40% w/v HA to obtain a porosity of > 85% and high compressive modality according to a previous report [30]. A mixture of PE with HA has been commonly used for orbital implants in humans. However, PE has the disadvantage of a high extrusion rate due to slow and incomplete fibro-vascular ingrowth [23, 40]. In contrast, HA, similar to bone minerals, favors fibro-vascular in-growth, thereby reducing the risk of extrusion, resisting infections, and facilitating the treatment of infection with antibiotics. Moreover, HA is biocompatible and nontoxic [41]. HA, with the general formula $Ca_{10}(PO_4)_6(OH)_2$, is an important mineral that exists in the body of humans and animals that accounts for most of the dry weight of bones and teeth, and gives stiffness to bones [42]. HA can fabricate diverse forms of scaffolds with porosities that are favorable for cell migration and attachment. However, because its brittle nature restricts load-bearing ability, HA is often mixed with other polymers to improve the function of implants [43].

PCL, a synthetic semicrystalline biodegradable polymer with good toughness, has been widely used as a biomaterial for scaffolds in tissue engineering [42]. However, it has poor strength [44–48]. To overcome these limitations, we mixed the PCL polymer with HA. This mixture not only resulted in a complementary effect by improving the strength and brittleness of the material, but also maximized the advantages of each material. A previous report showed that the mixture of PCL and HA has superior osteoconductivity and biocompatibility with low inflammation [49]. Moreover, because of its hydrophobicity, PCL has adverse effects on cell attachment and proliferation. To overcome these limitations, blending with other materials is needed [50]. Our results also show that mixture of PCL and HA was not only excellent to produce customized ocular implant but also biocompatible. Therefore, it could be applicable not only for animals but also human medicine, although further research is needed.

For printing ocular prosthesis, we used the PCL material. At present, PMMA prosthesis is generally used in human medicine. Because PMMA offers strength, plasticity, and lightweight, it has long been used as a biomedical material [51]. However, it can cause cellular damage and easy to infection. Moreover, in our preliminary study, we found that the PMMA filament was difficult to print due to its high melting point and high viscosity. In contrast, PCL has a low melting point and can be easily printed in 3D [44–46]. Moreover, it has excellent

biocompatibility, antibacterial effects, good toughness, flexibility, and no cytotoxicity. Its hydrophobic property allows it to withstand ocular discharge and regular washing [46]. Acrylic painting and light curing resin coating could be easily applied on the PCL prosthesis, without any noted deformity. The coating process softened the surface of the prosthesis. Additionally, it was biocompatible and did not cause tissue inflammatory reactions after *in vivo* application.

Pegs are used to connect orbital implants with ocular prosthesis, thus supporting the prosthesis and enhancing movement [6]. However, pegs were not used in this study because they may cause various complications such as chronic discharge, pyogenic granuloma formation, pain, and protrusion [52, 53]. As a result, the movement of the prosthesis was weak; however, in a dog eye, conjunctiva is almost invisible and only the cornea is mainly seen, when compared to a human eye. Therefore, it was well acceptable cosmetically.

In this study, we performed two kinds of eye removal surgical methods including evisceration and enucleation because they have been usually done in both human and animal clinics [54, 55]. First, evisceration is performed on unresponsive endophthalmitis or painful eyes with no vision. Evisceration removes the contents of the eye except the sclera, conjunctiva, Tenon's capsule, extraocular muscles and optic nerve. Second, enucleation removes the entire eyeball except conjunctiva, Tenon's capsule and extraocular muscles, which is performed mainly in severe trauma, intraocular tumor.

We noted mild to moderate ocular discharge after surgery. The main causes of decreased tear production may be lack of reflex tear secretion by corneal stimulation and resection of the nictitating membrane during surgery. Because the reflex tears provide most of the aqueous component, mucoid and lipid residues may remain after surgery [56]. In our study, use of lubricants or artificial tears may have improved this symptom. In fact, we removed the third eye lid during the eye removal surgery to ease the application of the ocular prosthesis and prevent the protrusion of the third eye lid [6]. The third eyelid acts as a wiper to remove debris and mucus, producing approximately one-third of the tear volume. Therefore, we recommend not to remove the nictitating membrane during surgery to increase the tear film area.

The limitation of the method developed in this study was that it still needs a manual process of coloring and coating the prosthesis. However, by replacing the process of creating an impression mold, a wax model, and an acrylic mold, the above mentioned method can greatly reduce the time and cost of production and increase the treatment efficacy. Although more *in vivo* clinical applications and longer follow-ups are warranted, the development of CMPs and ocular implants for companion animals using a 3D printing technology could have great potential in veterinary clinical practice.

## Conclusions

In this study, CMP and ocular implants for companion animals were developed using 3D printing technology, overcoming the limitations of a conventional eye removal surgery in dogs. In addition, the appearance of the artificial eye was good, and there were no significant complications associated with their use. Our 3D printed artificial eye may be beneficial for companion animals due to the reduction in the time and cost of manufacturing compared to the conventional methods.

## Supporting information

**S1 Fig. Evisceration method (4 petal technique) in a dog.** After performing 360° peritomy, sub-Tenon's blunt dissection and excision of the corneal button were done (A). Ocular contents were scooped out (B) and then four scleral petals were created (C). After inserting the ocular implant (D), the sclera was sutured (E). Two layers of scleral suture were provided for

additional cover of the implant. Tenon's capsule and conjunctiva were sutured respectively (F). The conformer was worn over the conjunctiva (G). Temporal tarsorrhaphy was maintained for 2 weeks (H).
(TIF)

**S2 Fig. Enucleation (myoconjunctival technique) method in a dog.** A 360˚ peritomy and sub-Tenon's blunt dissection were performed (A). After hooking the four rectus muscles (B), traction suture and stay suture were performed (C). Each of the rectus muscles was transected near its insertion (D). The eyeball was gently lifted using 4 traction sutures, and subsequently, the optic nerve was transected. The ocular implant was inserted (E). Each of the rectus muscles is sutured through the Tenon's capsule and conjunctiva using double-armed 6–0 polydioxanone. The Tenon's capsule and conjunctiva were sutured (F). The conformer was worn over the conjunctiva (G). Temporal tarsorrhaphy was maintained for 2 weeks (H).
(TIF)

**S1 Table. Setting conditions for 3D printing output of orbital implant and ocular prosthesis.**
(DOCX)

**S1 File.**
(DOCX)

## Author Contributions

**Conceptualization:** Kyung-Mee Park.

**Data curation:** So-Young Park, Jeong-Hee An.

**Formal analysis:** So-Young Park.

**Funding acquisition:** Kyung-Mee Park.

**Investigation:** So-Young Park, Jeong-Hee An, Hyun Kwon, Seo-Young Choi, Ka-Young Lim, Kyung-Mee Park.

**Methodology:** So-Young Park.

**Resources:** Kyung-Mee Park.

**Supervision:** Heung-Myong Woo, Kyung-Mee Park.

**Validation:** Kyung-Mee Park.

**Visualization:** So-Young Park, Kyung-Mee Park.

**Writing – original draft:** So-Young Park.

**Writing – review & editing:** Ho-Hyun Kwak, Kamal Hany Hussein, Heung-Myong Woo, Kyung-Mee Park.

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
