## [Decision Letter · Decision Letter 0]

15 Jul 2020

PONE-D-20-17365

Custom-made artificial eyes using 3D printing for dogs: a preliminary study

PLOS ONE

Dear Dr. Park,

Thank you for submitting your manuscript to PLOS ONE. After careful consideration, we feel that it has merit but does not fully meet PLOS ONE’s publication criteria as it currently stands. Therefore, we invite you to submit a revised version of the manuscript that addresses the points raised during the review process.

The reviewers granted a quick and constructive feedback which includes my fullest support and

I am looking forward to a revised version of your manuscript for further consideration.

We look forward to receiving your revised manuscript.

Kind regards,

Fabian Huettig, DMD, Ph.D.

Academic Editor

PLOS ONE

Journal Requirements:

1, Please ensure that your manuscript meets PLOS ONE's style requirements, including those for file naming. The PLOS ONE style templates can be found at

2.  Please indicate how often animal care staff monitored the health and well-being of the animals and the criteria used to make such assessments, and provide any additional details regarding housing conditions, feeding and exercise regimens, environmental enrichment if applicable. Thank you for your attention to this request"

3. We ask that you please consider moving Figures 1 and 2 to Supporting Information due to their graphic nature. Thank you for your consideration

Reviewers' comments:

Reviewer's Responses to Questions

**Comments to the Author**

1. Is the manuscript technically sound, and do the data support the conclusions?

Reviewer #1: Yes

Reviewer #2: Yes

2. Has the statistical analysis been performed appropriately and rigorously? 

Reviewer #1: N/A

Reviewer #2: N/A

3. Have the authors made all data underlying the findings in their manuscript fully available?

Reviewer #1: Yes

Reviewer #2: Yes

4. Is the manuscript presented in an intelligible fashion and written in standard English?

Reviewer #1: Yes

Reviewer #2: No

5. Review Comments to the Author

Reviewer #1: The idea of using 3D printing for making artificial eyes for dogs is a good one; the authors have demonstrated that it works.

People in several professions will be interested in this article. I suggest an amplification of Materials story in this paper. Why hydroxyapatite ? As discussed by W. Brostow and H.E. Hagg Lobland, Materials: Introduction and Applications, John Wiley & Sons 2017, hydroxyapatite is an important constituents of human and animal bodies; this deserves to be said. Also, similarly the authors can explain the reasons for using polycaprolactone vs. PMMA.

Reviewer #2: Dear editor and authors, thank you for the opportunity offered to me to review this work on the digital workflow in veterinary rehabilitation.

The authors are to be commended for their innovative work, as they sought to integrate the digital protocol into a daily practice of animals eye enucleation treatment.

Nonetheless, as it is, the manuscript presents some major weaknesses that should preclude its publication:

1). I must state that the manuscript was not easy to read and the described technical approach was hard to follow. This fact is surely attributed to the weak structure of this article, as the Materials&Methods and Results sections are a kind of combined together.

For instance, you describe the digital data acquisition and go like:

The size of the orbital implant was made to be 75% of the ocular volume by measuring the longest diameter of the eyeball using B-mode ultrasonography, according to a previous report [26]. For size of ocular prosthesis, the X-axis and Z-axis were determined by the diameter

For me this belong surely to the M&M Section

So I suggest that you should clearly state what was done in the M&M section touching upon the following points:

-clinical examination

-Surgical approaches and post-op phase

-digital data acquisition

-CAD

-CAM

Then you switch to RESULTS and describe:

-the prosthesis fit

-dogs owns perception

-time and money investments

2). As the for the structure in the Introduction section, I suggest you should change it as well. In the middle of the Introduction section you state the aim of this study, and the start to describe the benefit of 3D printing utilization and its application in human facial prosthetics. I suggest the aim should be stated clearly and goes to the end of the section.

You go like:

Herein, we aimed to create custom-made artificial eyes including ocular implants and

75 prosthesis for dogs using 3D printing in order to overcome these challenges

I suggest the aim of this clinical trial was not only to create the 3D printed eye prosthese, but to also to evaluate their clinical performance on two dogs cases and, what was in fact done.

So please, do rephrase the aim and once again check the introduction section.

3) Currently, custom-made prostheses (CMPs) made of PMMA are mainly used in humans because they are more comfortable and esthetically more acceptable than ready-made prosthesis.

Pleaser support this statement with a citation.

4) Currently, 3D printed bio-models are widely used in the medical field because they can provide tactile feedback and reproduce anatomical structures and movements.

You need to provide a strong citation for this statement.

5) The 3D printing technology is used not only for planning surgical procedures, making intraoperative guidance devices, and training but also for producing facial prostheses of the nose and ear [23-25].

The citations 23 and 25 seem to be the wrong one, as they don’t relate neither to nose nor ear prostheses. Please check and consider citing for instance: PMID: 29145528 or PMID: 32611482

6) 367 Previously, a report scanned the wax model 367 using computed tomography (CT)

I can not follow this sentence; please rephrase.

7) Please consider describing the other studies in impersonal way, avoiding such pronouns as he, they. Use the passive voice instead.

8) Why did you perform the both surgical evisceration and enucleation approaches? Was their any relevance for this?

9) Please do a spell check

6. PLOS authors have the option to publish the peer review history of their article (what does this mean?). If published, this will include your full peer review and any attached files.

Reviewer #1: **Yes: **Witold Brostow

Reviewer #2: **Yes: **Alexey Unkovskiy

---

## [Author Response · Author response to Decision Letter 0]

8 Sep 2020

Journal Requirements:

1, Please ensure that your manuscript meets PLOS ONE's style requirements, including those for file naming. The PLOS ONE style templates can be found at

>> We checked and followed PLOS ONE's style requirements.

2. Please indicate how often animal care staff monitored the health and well-being of the animals and the criteria used to make such assessments, and provide any additional details regarding housing conditions, feeding and exercise regimens, environmental enrichment if applicable. Thank you for your attention to this request"

>> Line 142-150: We included the required information of animal experiments in our manuscript.

3. We ask that you please consider moving Figures 1 and 2 to Supporting Information due to their graphic nature. Thank you for your consideration

>> We moved Figures 1 and 2 to Supporting Information (S1 and S2 Fig). 

Reviewers' comments:

Reviewer's Responses to Questions

5. Review Comments to the Author

Reviewer #1: The idea of using 3D printing for making artificial eyes for dogs is a good one; the authors have demonstrated that it works.

Q. People in several professions will be interested in this article. I suggest an amplification of the Materials story in this paper. Why hydroxyapatite ? As discussed by W. Brostow and H.E. Hagg Lobland, Materials: Introduction and Applications, John Wiley & Sons 2017, hydroxyapatite is an important constituents of human and animal bodies; this deserves to be said. Similarly, the authors can explain the reasons for using polycaprolactone vs. PMMA.

>> Thank you for the reviewing our manuscript.

>> In response to the reviewer’s comment, we have improved the material stories in our manuscript. 

>> Line 391-398, 401-405, 409-411

Reviewer #2: Dear Editor and authors, thank you for the opportunity offered to me to review this work on the digital workflow in veterinary rehabilitation.

The authors are to be commended for their innovative work, as they sought to integrate the digital protocol into a daily practice of animal eye enucleation treatment.

Nonetheless, as it is, the manuscript presents some major weaknesses that should preclude its publication:

1). I must state that the manuscript was not easy to read and the described technical approach was hard to follow. This fact is surely attributed to the weak structure of this article, as the Materials&Methods and Results sections are a kind of combination.

For instance, you describe the digital data acquisition and go like:

The size of the orbital implant was made to be 75% of the ocular volume by measuring the longest diameter of the eyeball using B-mode ultrasonography, according to a previous report [26]. For size of ocular prosthesis, the X-axis and Z-axis were determined by the diameter. 

For me, this belongs surely to the M&M Section

So I suggest that you should clearly state what was done in the M&M section touching upon the following points:

clinical examination

-Surgical approaches and post-op phase

-digital data acquisition

-CAD

-CAM

Then you switch to RESULTS and describe:

-the prosthesis fit

-dogs owns perception

-time and money investments. 

>> Thank you for the reviewing our manuscript.

>> In response to the reviewer’s comment, we have revised our M&M and Results sections to describe these points. Please refer to these sections in our manuscript.

>> Line 87, 105, 155, 205, 249, 299, 313, 330

2). As for the structure in the Introduction section, I suggest you should change it as well. In the middle of the Introduction section you state the aim of this study, and the start to describe the benefit of 3D printing utilization and its application in human facial prosthetics. I suggest that the aim should be stated clearly and goes to the end of the section.

You go like:

Herein, we aimed to create custom-made artificial eyes including ocular implants and

75 prostheses for dogs using 3D printing in order to overcome these challenges. 

I suggest the aim of this clinical trial was not only to create the 3D printed eye prosthese, but also to evaluate their clinical performance in two dog cases and, what was in fact done. Therefore, please rephrase the aim and once again check the introduction section.

>> In response to the reviewer’s comment, we have revised our Introduction. The aim was rephrased and provided at the end of the section.

>> Line 80-82.

3) Currently, custom-made prostheses (CMPs) made of PMMA are mainly used in humans because they are more comfortable and esthetically more acceptable than ready-made prostheses.

Pleasers support this statement with a citation.

>> Line 66-68: In response to the reviewer’s comment, we have added the citation. 

4) Currently, 3D printed bio-models are widely used in the medical field because they can provide tactile feedback and reproduce anatomical structures and movements.

You need to provide a strong citation for this statement.

>> Line 76-78: In response to the reviewer’s comment, we have added the citations.

5) The 3D printing technology is used not only for planning surgical procedures, making intraoperative guidance devices, and training, but also for producing facial prostheses of the nose and ear [23-25].

 Citations 23 and 25 seem to be the wrong ones, as they do not relate either to nose or ear prostheses. Please check and consider citing for instance: PMID: 29145528 or PMID: 32611482

>> Line 78-80: In response to the reviewer’s comment, we have corrected these specific citations. 

6) 367 Previously, a report scanned the wax model 367 using computed tomography (CT)

I cannot follow this sentence; please rephrase.

>> In response to the reviewer’s comment, we have rephrased the sentence.

>> Line 365-366 : In a previous report, the wax model was produced by computed tomography (CT) scanning, and then the prosthesis was fabricated by 3D printing 

7) Please consider describing the other studies in an impersonal way, avoiding such pronouns as he, they. Use the passive voice instead.

>> In response to the reviewer’s comment, we have rephrased our sentences.

>> Line 371-373: To overcome these limitations, a prosthesis was created by scanning the anophthalmic socket using CT, without creating an impression mold in the conventional way [1].

>> Line 378-379: A digital light processing (DLP) printer with liquid materials was used with high precision and output. 

8) Why did you perform both surgical evisceration and enucleation approaches? Was any relevance for this?

>> We have added an explanation for this in the manuscript.

>> Line 425- 431

In this study, we performed two kinds of eye removal surgical methods including evisceration and enucleation because they have been usually done in both human and animal clinics. First, evisceration is performed on unresponsive endophthalmitis or painful eyes with no vision. Evisceration removes the contents of the eye except the sclera, conjunctiva, Tenon’s capsule, extraocular muscles and optic nerve. Second, enucleation removes the entire eyeball except conjunctiva, Tenon’s capsule and extraocular muscles, which is performed mainly in severe trauma, intraocular tumor. 

9) Please do a spell check

>> In response to the reviewer's comment, our manuscript has been reviewed by a native English speaker.

---

## [Decision Letter · Decision Letter 1]

23 Sep 2020

PONE-D-20-17365R1

Custom-made artificial eyes using 3D printing for dogs: a preliminary study

PLOS ONE

Dear Dr. Park,

Thank you for submitting your manuscript to PLOS ONE. After careful consideration, we feel that it has merit but does not fully meet PLOS ONE’s publication criteria as it currently stands. Therefore, we invite you to submit a revised version of the manuscript that addresses the points raised during the review process.

Could you please fix the minor issues which have been raised by the reviewers

including a thorough textpolishing -

and I am looking forward to crosscheck your corrections for final approval.

We look forward to receiving your revised manuscript.

Kind regards,

Fabian Huettig, DMD, Ph.D.

Academic Editor

PLOS ONE

Additional Editor Comments (if provided):

Could you please fix the minor issues which have been raised by the reviewers

including a thorough textpolishing -

and I am looking forward to crosscheck your corrections for final approval.

Reviewers' comments:

Reviewer's Responses to Questions

**Comments to the Author**

1. If the authors have adequately addressed your comments raised in a previous round of review and you feel that this manuscript is now acceptable for publication, you may indicate that here to bypass the “Comments to the Author” section, enter your conflict of interest statement in the “Confidential to Editor” section, and submit your "Accept" recommendation.

Reviewer #1: (No Response)

Reviewer #2: All comments have been addressed

2. Is the manuscript technically sound, and do the data support the conclusions?

Reviewer #1: Yes

Reviewer #2: Yes

3. Has the statistical analysis been performed appropriately and rigorously? 

Reviewer #1: N/A

Reviewer #2: N/A

4. Have the authors made all data underlying the findings in their manuscript fully available?

Reviewer #1: Yes

Reviewer #2: Yes

5. Is the manuscript presented in an intelligible fashion and written in standard English?

Reviewer #1: Yes

Reviewer #2: No

6. Review Comments to the Author

Reviewer #1: I have found an error in English in a sentence "KM Park make the experimental ideas", so please correct.

In Ref. 42 the name of the second author should be Hagg Lobland HE, the year 2017.

Reviewer #2: Dear Authors, thank you for your respond.

I would like to adress some minor revisions:

Line 330: "No clinical complications such as inflammation, infection, or extrusion of the ocular implant observed during the 6 months after surgery"

The verb "were" is missing

Line 69

superior a esthetics. This does not sound good. Please rephrase.

Line 354 "We found that using 3D printing, the production time was shortened, the price was lowered, and the labor required was reduced for the production of the artificial eyes"

Rephrase the sentence as follows "We found that application of 3D printing allows to decrease the production time...."

LIne 364 "In a previous report, wax model was produced by computed tomography (CT) scanning and then fabricated the prosthesis by 3D printing"

Rephrase the sentence and omit the word "scanning", as it is not applicable with regards to CT.

LIne 369 In addition, the soft tissue distortion of the anophthalmic socket could occur owing to the pressure of the molding material during the process of obtaining the impression mold, and an incorrect mold may be obtained.

Please omit the last pasrt of the sentence. ("during the process of obtaining the impression mold, and an incorrect mold may be obtained").

Line 376 More recently, prostheses was produced using 3D printing more accurately. Either "Prosthesis was" or "Prostheses were"

LIne 442 "the abovementioned" split these two words

LIne 449. Conclusion must be rephrased. Please avoid using the pronouns like "We" and use the passive voice.

7. PLOS authors have the option to publish the peer review history of their article (what does this mean?). If published, this will include your full peer review and any attached files.

Reviewer #1: No

Reviewer #2: No

---

## [Author Response · Author response to Decision Letter 1]

18 Oct 2020

Response to reviewers

Reviewer #1: 

Q1. I have found an error in English in a sentence "KM Park make the experimental ideas", so please correct.

A> Thank you for the reviewing our manuscript.

We corrected the sentence. 

“KM Park designed the experiments.” (Line 464)

Q2. In Ref. 42 the name of the second author should be Hagg Lobland HE, the year 2017.

A> We corrected the author name. 

“42. Brostow W, Hagg Lobland HE. Materials: introduction and applications: John Wiley & Sons; 2017.” (Line 507)

Reviewer #2: 

Q1. I would like to adress some minor revisions:

Line 330: "No clinical complications such as inflammation, infection, or extrusion of the ocular implant observed during the 6 months after surgery"

The verb "were" is missing

A> Thank you for the reviewing our manuscript. 

We corrected the sentence. 

“No clinical complications such as inflammation, infection, or extrusion of the ocular implant were observed during the 6 months after surgery (Fig.6A, a).” (Line 330-331)

Q2. Line 69 superior a esthetics. This does not sound good. Please rephrase.

A> We rephrased the sentence. 

“Their use in small animals is relatively uncommon although they have shown good results, with few major complications and better appearance.” (Line 69-70)

Q3. Line 354 "We found that using 3D printing, the production time was shortened, the price was lowered, and the labor required was reduced for the production of the artificial eyes" Rephrase the sentence as follows "We found that application of 3D printing allows to decrease the production time...."

A> We rephrased the sentence. 

“We found that application of 3D printing allows to decrease the production time, price, and the labor for the production of the artificial eyes” (Line 355-356)

Q4. LIne 364 "In a previous report, wax model was produced by computed tomography (CT) scanning and then fabricated the prosthesis by 3D printing" Rephrase the sentence and omit the word "scanning", as it is not applicable with regards to CT.

A> We rephrased the sentence. 

“In a previous report, wax model was produced by computed tomography (CT) and then fabricated the prosthesis by 3D printing” (Line365-366)

Q5. LIne 369 In addition, the soft tissue distortion of the anophthalmic socket could occur owing to the pressure of the molding material during the process of obtaining the impression mold, and an incorrect mold may be obtained. Please omit the last pasrt of the sentence. ("during the process of obtaining the impression mold, and an incorrect mold may be obtained").

A> We rephrased the sentence.

“In addition, the soft tissue distortion of the anophthalmic socket could occur owing to the pressure of the molding material” (Line 369-370)

Q6. Line 376 More recently, prostheses was produced using 3D printing more accurately. Either "Prosthesis was" or "Prostheses were"

A> We corrected the sentence. “prostheses were” (Line 376)

Q7. LIne 442 "the abovementioned" split these two words

A> We corrected the sentence. “the above mentioned” (Line 442)

Q8. line 449. Conclusion must be rephrased. Please avoid using the pronouns like "We" and use the passive voice.

A> We rephrased the sentence.

“In this study, CMP and ocular implants for companion animals were developed using 3D printing technology, overcoming the limitations of a conventional eye removal surgery in dogs.” (Line 449-450)

---

## [Editor Report · Decision Letter 2]

30 Oct 2020

Custom-made artificial eyes using 3D printing for dogs: a preliminary study

PONE-D-20-17365R2

Dear Dr. Park,

We’re pleased to inform you that your manuscript has been judged scientifically suitable for publication and will be formally accepted for publication once it meets all outstanding technical requirements.

Kind regards,

Fabian Huettig, DMD, Ph.D.

Academic Editor

PLOS ONE

Additional Editor Comments (optional):

Thank you for your corrections and improvements to the manuscript.

I do apologize the delay in processing and hope that your research will

be available ASAP to the scientific community within PLOS ONE.